# Social Workers' Involvement in Policy Practice in Portugal

**Rita Carrilho** [1] and **Francisco Branco** [2,*]

1    ISCTE-Instituto Universitário de Lisboa, Avenida das Forças Armadas, 1649-026 Lisboa, Portugal
2    Católica Research Centre for Psychological, Family and Social Wellbeing, Universidade Católica Portuguesa, Palma de Cima, 1649-023 Lisboa, Portugal
*    Correspondence: fnbranco@ucp.pt

**Abstract:** This article presents the results of the first survey-based study in Portugal about the level of involvement of social workers in social policies, aiming to determine if policy practice is embedded in Portuguese social workers' professional practice and which factors may enhance or constraint such practice. Combining the Civic Voluntarism Model by Verba and colleagues and the Policy Practice Engagement Model by Gal and Weiss-Gal, which were considered as the main predictors of social workers' engagement in policy practice, this study followed a quantitative approach, based on 265 valid answers to an online survey obtained through snowball sampling. The findings showed a low level of engagement in policy practice activities among the social workers, especially those requiring a greater public exposure and acting with the media, policymakers, or public officers to share opinions, make a proposition, or report a problem. Considering the main factors of the socio-political context, the professional context, the organisational context, and individual factors, the findings showed that individual factors explained most of the differences in the social workers' involvement in policy practice, especially when considering interest and efficacy. This study pointed out the need for further research in this area.

**Keywords:** social work; social policy; policy practice; Civic Voluntarism Model; Policy Practice Engagement Model; Portugal





## 1. Introduction

Social work and social policy are closely interlinked. As Morales and Sheafor (2002) emphasise, "social work did not evolve in a vacuum" because its professional practice is influenced by the "decisions about the extent to which [the] society would respond to its members' social needs and, subsequently, to the social programs that would be supported." (p. 51). According to this lens, public policies, as collective responses to socially recognised problems and needs, constitute one of the essential frameworks of social workers' professional practice and, therefore, a structuring dimension for the exercise of the profession through policies of social security, social protection policies, and other public programmes, as well as policies affecting the living conditions of users and services, such as education, health, and housing, and policies concerning rights, citizenship, and democracy itself (Solender 1958)

The policy cycle encloses spaces that allow social workers to constitute themselves as actors of public policies and exert their influence at different stages of the policymaking process, not only in the "implementation game" (Knoepfel et al. 2011), but also in the other phases of the policy cycle [problem (re)definition and agenda setting, formulation, and decision making] (Klammer et al. 2020).

Social work professionals can play a significant role in policy agenda setting by identifying relevant social problems and calling for people's, media's, and policymakers' attention. They may also play an active role in the policymaking process, seeking policies that are formulated in line with the problems they are intended to solve. Given their experience and empirical knowledge in dealing with legislation and its limitations, social

workers can contribute to identify what works and what does not in social policy measures and programmes. They may also provide valuable insights and knowledge for programme evaluations and policy improvements.

Social workers are close to individuals and their social contexts, and they are often in the first line of social problem responses. Therefore, as Mendes (2007) points out, "social workers can transform private pain into public issues" (p. 41).

Following this perspective, policy practice can be described as the set of social workers' professional activities developed to bring service users' individual problems to the public sphere. Such practice may provide input to policies from the "bottom-up". In this sense, the reinforcement of policy practice has a double-edged dimension: The first is the reinforcement of the social role of social work, through the recovery of the core professional values that lie at the emergency and institutionalisation of the social work profession. The second is the recommitment of social policies to the furthering of social wellbeing and justice (Figueira-McDonough 1993; Jansson 2008; Gal and Weiss-Gal 2013; Amaro 2015).

The role reserved for social work in the public policy process is often limited to policy implementation, i.e., putting legislation and social programmes into practice. This phase of the policy cycle incorporates several opportunities for professionals to act. The individual freedom with which a professional interprets a legislation, according to ethical-professional principles, and the level of discretion in decision making about social provisions and social services, which is provided by organisational and institutional contexts and has different sets of rules and structures of authority, particularly in front-line welfare services (Evans 2011; Evans and Harris 2004; Lipsky [1980] 2010), give these professionals a margin of action beyond strict regulation enforcement.

As Gal and Weiss-Gal and Weiss-Gal (2013, Introduction) argue, the "[T]he history of the social work profession is replete with examples of social workers seeking to influence policies in the societies in which they lived" (p. 1), but this tradition has not precluded the fact that, until recent years, the social work discourse and practice "tend to delegate the role of involvement in social policy to a small group of social policy experts and to community social workers"(p. 4).

Policy practice in social work challenges this view of the division between social workers and social policy experts. The term "policy practice", first coined by Jansson in 1984, has been particularly developed in the USA in the 1980s and 1990s (e.g., Figueira-McDonough 1993; Jansson 1984; Wyers 1991), and in recent decades, it has gained increasing interest in the social work literature with social work and other academic scholars pursuing several lines of research, such as studies on the ways of involvement in policy practice (Pawar 2019; Saxena and Chandrapal 2022; Weiss-Gal et al. 2020); factors enhancing or constraining policy practice (De Corte and Roose 2020; Weiss-Gal and Gal 2018); social work education for policy practice (Balaz 2022; Pawar and Nixon 2020; Pritzker and Lane 2014; Weiss-Gal and Gal 2019; Weiss et al. 2006; Zubrzycki and McArthur 2004); academic involvement in policy practice (Gal and Weiss-Gal and Weiss-Gal 2017); societal and welfare state changes impacting on social work professional activity and policy-related activities (Klammer et al. 2020; Strier and Feldman 2018); social work policy engagement both at the macro and micro levels and fields (Elmaliach-Mankita et al. 2019; Gal and Weiss-Gal and Weiss-Gal 2020; Sery and Weiss-Gal 2022; Weiss-Gal 2017); the role of social work associations in policy making (Guidi 2020); and social workers' political participation (Ritter 2007, 2008; Rome and Hoechstetter 2010) or policy engagement through running for an elected office (Amann and Kindler 2022; Binder and Weiss-Gal 2022; Lane 2011; Leitner and Stolz 2022).

In this article, following Gal and Weiss-Gal and Weiss-Gal (2013), policy practice is defined "as an integral part of their professional activity in diverse fields and types of practice, that focus on the formulation and implementation of new policies, as well as on existing policies and suggested changes in them" (pp. 4–5). In other words, policy practice refers to social workers' professional policy-related activities and not to their political activities as citizens. This definition was reformulated in the Policy Practice Engagement

Conceptual Framework (PPE) (cf. Gal and Weiss-Gal 2015) and, as detailed in Section 2, was adopted in this research.

This article addresses the relationship between social work and social policy in Portugal and is based on the first study to examine if Portuguese social workers incorporate activities aimed at influencing social policies (policy practice) in their professional routines, which factors enhance or constraint such practice, and what is the level of involvement of social workers in policy practice in Portugal. Our goal was to ascertain if social workers in Portugal engage in policy practice activities, which activities show the highest level of engagement, and which factors enhance or constraint such engagement.

The research design followed similar studies conducted in other countries (Ritter 2008; Hoefer 2013; Gal and Weiss-Gal 2013; Gewirtz-Meydan et al. 2016).

The results lead to the conclusion that the inquired social workers show a low level of engagement in policy practice activities, especially in activities requiring a greater public exposure, such as using the media or contacting policymakers or public officers to voice an opinion, present a proposal, or report a problem. Th study results reveal that engaging in policy practice depends mostly on individual factors, mainly social workers' personal interests in social policy matters and their perception of being able to "make a difference".

The results point to Portuguese social workers acknowledging a strong connection between social work and social policies although they perceive that policies are not providing enough answers to people's problems, creating low levels of satisfaction. They also acknowledge that there is room for social workers to enter the public policy arena, but they are not conducting their professional activities in that direction.

Policy practice is an underexplored dimension of social work professional practice in Portugal, and based on this study, we outlined some strategies for reinforcing such practice and academic research in this field.

## 2. Conceptual Model and Methods

The conceptual model of this study is based on the Civic Voluntarism Model (Verba et al. 1995) on political participation and the Policy Practice Engagement Model (Gal and Weiss-Gal 2015). The Civic Voluntarism Model (CVM) addresses the question of why citizens do not participate in political processes. Verba and his colleagues concluded that there are three main inhibiting factors for political participation: (1) the lack of resources (monetary, educational, and time), (2) the lack of motivation (low interest, concern, and knowledge or sense of efficacy), and (3) the lack of opportunity (invitations in personal networks). In sum, citizens engage in political participation activities when they can (resources), when they want to (motivation), or when somebody asks them to (opportunity) (Verba et al. 1995). Following (Verba et al. 1995; Norris 2002) distinguishes a structural macro level concerning socio-economic development and state structure; a meso level concerning institutions, such as non-profit organisations, political parties, trade unions, and religious organisations; and a micro level concerning individual's resources and sources of motivation, such as interest, perception of being able to make a difference, and civic and educational skills. While the CVM addresses political participation of individuals, the Policy Practice Engagement Model (PPE) addresses social workers' policy participation as part of their professional capacity. According to this analytical model, social workers' engagement in policy practice activities depends on the opportunities offered by the socio-political context (policy process), the facilitation that the professional context and the organisational setting offer, and the motivation influenced by individual factors. The opportunity component regards the socio-political context, which can be described as the policy process institutional setting and the degree to which social workers can access it. The facilitation component concerns the professional context. The organisational culture, which is understood as the dominant values, norms, and expected modes of behaviour within a specific workplace, may promote or discourage activities related to policy practice engagement. The motivation component regards individual choices based on values, attitudes, and capacities, along with external and internal sources. External sources include

social work education, professional association activities, and professional discourses, and internal sources are related to an individual's personal background, values, and interests (Gal and Weiss-Gal 2015).

The adaptation of the PPE model in this study (Figure 1) considered that the level of policy practice engagement could be explained by the macro-level socio-political context, the meso-level professional context (as a facilitator), and the micro-level individual factors (as motivation drivers). The socio-political and professional contexts and the organisational setting are external to individuals in the sense that they exist despite the individuals and do not result from individual choices. According to this assumption, in the same socio-political or professional contexts, some social workers can decide to engage, and others can decide the other way around. For this reason, external sources of motivation, such as social work education or the role of professional associations and discourses, which are taken as individual factors in the PPE model, were included in the present study in the facilitation component as being external to individuals. The results section provides a detailed explanation of the dimensions considered in each component of the analytical model summarized in Figure 1.

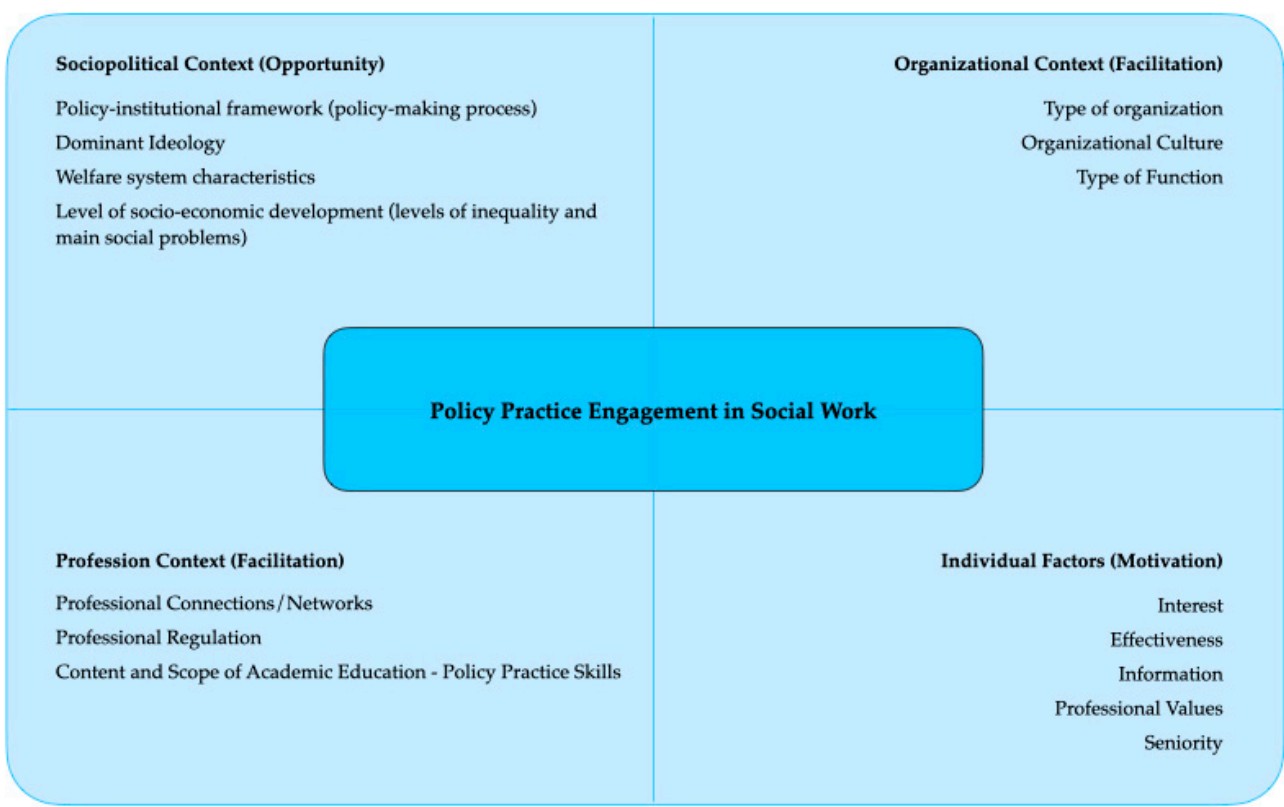

**Figure 1.** Policy Practice Analytical Model. Source: Carrilho (2018)'s adaptation of the Civic Voluntarism Model (Verba et al. 1995) and the Policy Practice Engagement Model (Gal and Weiss-Gal 2015).

This research was based on a quantitative approach. The data was gathered using a snowball sampling method and collected through an online survey to social workers in Portugal, using the Qualtrics platform. To avoid responses from other professionals, a question was introduced to ascertain the respondents' academic training (only professionals with an academic degree in social work were considered). The data were collected during the last quarter of 2017. There were 447 answers, but only 265 (59%) were considered valid answers because some of the respondents were excluded since they did not answer all the survey's mandatory questions. As mentioned, the data were collected through a snowball sampling method, given that there is no official database for all social workers in Portugal and, in this circumstance, it is not possible to build a randomized and/or stratified sample. The survey was disseminated to members of the Portuguese association

of social workers by official mailing, via social media, and by contacting active online groups of social workers, and respondents were asked to share the survey with other licensed social workers. We estimated a 20,609 social workers' population in 2016 in Portugal by updating the estimates made by Branco (2009) of 14,875 people with a social degree until 2008 and adding 14,875 people with a social work degree from 2009 until 2016. The number of people with a degree in social work was used as a proxy for the number of active social workers, although it should be acknowledged that not everyone with a social work degree works, or is still active, as a social worker. Based on these estimates, a statistically significant sample would require at least 378 valid answers. Given the sampling limitations—snowball collection, unknown population, and low number of responses—our results should be considered with caution since our sample wasn't representative of the entire population of social workers. The survey aimed at practising social workers, with a set of 42 questions adapted for the Portuguese context from similar surveys on social workers' political participation and professional practice aiming to influence social policies (Gewirtz-Meydan et al. 2016; Ritter 2008) and from the European Social Survey (ESS round 8, 2016); the survey was constructed to simultaneously benefit from the experience and knowledge of previous studies and to contribute with data for cross-country analysis. There were 3 questions about the socio-political context, 5 questions about the organisational context, 5 questions about the profession context, and 17 questions about individual characteristics. Twelve questions for socio-demographic characterization were also included (Carrilho 2018). A total of 92.7% of the respondents are female, with an average age of 41.8 years old and a professional seniority profile of, on average, 15 years of professional experience. The data analysis was conducted during the first quarter of 2018.

## 3. Results

### 3.1. Sample's Sociodemographic Profile

With regard to gender distribution, most respondents (92.7%) stated that they are female. Although we do not have a census base, this distribution confirms the high feminization of the social work profession (cf. Branco 2009).

The ages of the respondents range from 22 to 71 years of age, with the average age being 41.8 years old. Considering the distribution by age bracket, the highest number of answers is concentrated in the age bracket of 36 to 45 years old. We can see that 65.9% of the respondents are up to 45 years old and 88% are up to 55 years old. This profile indicates that most of the respondents are in the middle stage of their professional career.

The responses obtained are spread across 17 of the 18 districts in mainland Portugal and the autonomous regions. Lisbon was the district with the highest number of answers (39%), followed by the most populated districts (Porto, Braga, and Leiria e Setúbal). Thus, the respondents cover all districts and regions of the country, except for Bragança.

Regarding academic qualifications, about 48% of respondents held a bachelor's degree, and of 51% of respondents held post-graduate academic qualifications (mba/postgraduation, master's, and doctorate) (cf. Table 1).

**Table 1.** Sample's Sociodemographic Profile.

| Dimension | | Count | Valid % |
|---|---|---|---|
| | Female | 243 | 92.7% |
| | Male | 19 | 7.3% |
| Gender Distribution | System Missing | 3 | |
| | Total | 265 | 100.0% |

**Table 1.** *Cont.*

| Dimension | | Count | Valid % |
|---|---|---|---|
| Age Distribution | Up to 25 years old | 21 | 8.1% |
| | From 26 to 35 years old | 57 | 22.1% |
| | From 36 to 45 years old | 92 | 35.7% |
| | From 46 to 55 years old | 57 | 22.1% |
| | From 56 to 65 years old | 26 | 10.1% |
| | Over 66 years old | 5 | 1.9% |
| | System Missing | 7 | |
| | Total | 265 | 100.0% |
| Academic Qualification | Bachelor | 124 | 48.1% |
| | MBA/Postgraduation | 56 | 21.7% |
| | Master | 61 | 23.6% |
| | Doctorate | 15 | 5.8% |
| | Other | 2 | 0.8% |
| | System Missing | 7 | |
| | Total | 265 | 100.0% |

### 3.2. Social Work and Social Policy relationship

The socio-political context respects to the functioning of the political system and the policy process, the dominant ideology, the welfare regime, and the socio-economic development level of a country. Variations in this context provide different opportunities for influencing the policy process (Gal and Weiss-Gal 2015). For this dimension, we considered it relevant to ascertain if social workers feel that they have the opportunity to participate in the policy process, if they are satisfied with the Portuguese welfare regime, and how they assess social problems affecting citizens.

The study results reveal a detachment between social workers and the political sphere, given the low levels of trust in the main political institutions, the unsatisfaction with social policies and the welfare system, the low levels of political participation, and the perception that social worker's opinions have little value for policymakers. This detachment may present an obstacle for engaging in policy practice activities which, by definition, imply entering the policy arena.

The results also reveal a general dissatisfaction with the Portuguese welfare system. A total of 82% of the respondents consider that social policies are inadequate, and 76.2% are unsatisfied with social policies. These results provide evidence of a contradiction between the expected role of social workers and the actual results they can achieve.

The inquired social workers also think that politicians do not take their opinions into account (76.2% consider that politicians worry very little or nothing). When it comes to political participation, the results show a low level of political participation, either in civic or social movements or in political parties. Despite the low level of political participation, social workers give much importance to social worker organisations as a professional group (91.3%). The gap between the discourse on the profession and the professional practice noticed in other studies (Gal and Weiss-Gal 2013) is also present in this study. In social work organisations' statements on professional mission, role, and values, policy practice is a constitutive dimension of the social work profession. From a professional practice perspective, policy practice has low significance. The inquired social workers consider that it is within social work's framework influencing policies (92.9%) suggesting there are no ethical constraints in engaging in policy practice. Even though they acknowledge their potential role in influencing social policies, the respondents reveal that they do not have

the required skills to do so, especially when addressing policymakers or the media (only 17% claim they are well or very well prepared to try and influence politics) (cf. Table 2).

**Table 2.** Respondents' skills in policy practice.

| Skills in PP | Preparation | Count | Valid % |
|---|---|---|---|
| Understand the role of social workers in social policy formulation | No preparation | 3 | 1.1% |
| | Low preparation | 28 | 10.6% |
| | Some preparation | 83 | 31.3% |
| | Good preparation | 119 | 44.9% |
| | Excellent preparation | 32 | 12.1% |
| | Total | 265 | 100.0% |
| Try to influence social policy decisionmakers | No preparation | 19 | 7.2% |
| | Low preparation | 64 | 24.2% |
| | Some preparation | 94 | 35.5% |
| | Good preparation | 75 | 28.3% |
| | Excellent preparation | 13 | 4.9% |
| | Total | 265 | 100.0% |
| Go to the media to call for attention to social problems | No preparation | 55 | 20.8% |
| | Low preparation | 82 | 30.9% |
| | Some preparation | 83 | 31.3% |
| | Good preparation | 39 | 14.7% |
| | Excellent preparation | 6 | 2.3% |
| | Total | 265 | 100.0% |
| Motivate service users to engage in actions to improve social policies | No preparation | 16 | 6.1% |
| | Low preparation | 33 | 12.5% |
| | Some preparation | 84 | 31.8% |
| | Good preparation | 107 | 40.5% |
| | Excellent preparation | 24 | 9.1% |
| | Total | 264 | 100.0% |

*3.3. Policy Practice Engagement*

3.3.1. Policy Practice Activities

Policy practice engagement (PPE) was measured using a set of 15 items describing activities to which the respondents answered "Yes" or "No". These items were adapted from the works of different authors (Verba et al. 1995; Ritter 2008; Figueira-McDonough 1993; Gewirtz-Meydan et al. 2016), and described various activities, such as contacting policymakers, celebrities, or the media; taking part in professional meetings and workgroups addressing social policy issues; encouraging service users or other social workers to take some action; and taking part in protest activities, amongst others.

The most frequent activities are those that involve discussing social problems and policies with other social workers, and the least frequent activities are those that are somehow related to protest activities or that require acting in the public sphere and the political arena, such as contacting the media or policymakers cf. Figure 2). These results suggest that Portuguese social workers prefer acting in the backstage to bring some influence on social policies, avoiding public exposure. Such preference may relate to the low policy practice skills acquired during academic training (cf. Table 2) as well as the already mentioned detachment from political institutions.

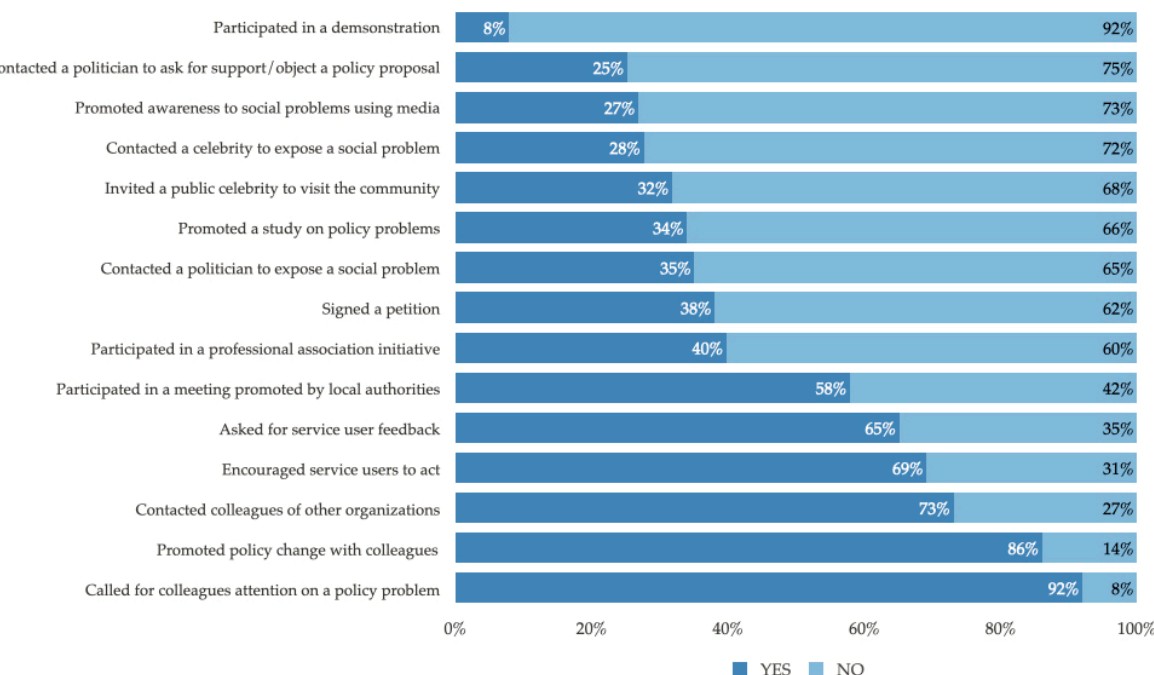

**Figure 2.** Engagement in PP activities in the past 12 months (%).

### 3.3.2. Level of Policy Practice Engagement

The level of engagement in policy practice was measured using a scale ranging from zero (answering "no" to all activities) to one (answering "yes" to all activities). We found a low level of policy practice engagement (mean = 0.47, $\alpha$ = 0.75), which is in line with similar studies conducted in other countries (Gal and Weiss-Gal 2013, 2015).

### 3.3.3. Factors That May Enhance or Constraint Policy Practice

For the socio-political context, the main predictor is "trust". It was measured using a 7-item scale, ranging from 1 (no trust at all) to 5 (total trust), showing a low level of trust in political institutions of 2.8 ($\alpha$ = 0.77). The lowest level of trust is directed to the most important political institutions in the policy process (Parliament, Government, and political parties), and the most trusted are social economy institutions.

For the professional context, we focused on policy practice skills. Ranging from one (no skills at all) to five (excellent skills), this scale was composed of four items, asking if social workers consider that they have acquired the necessary skills in their academic training to relate social work and social policy formulation, to try to influence policymakers, to contact the media, and to motivate service users to act for better social services. We found an average value of 3.08 ($\alpha$ = 0.829).

Only 9.1% of the respondents are involved in activities related to political decisions, while 73.9% are involved in the implementation of policies, dealing with caseworks, confirming the top-down relationship between social policy and social work. We could not, however, establish a relation between these results and the level of policy practice engagement shown. We could not find a consistent scale for the organisational setting, so we could not show how it may enhance or constraint policy practice engagement.

For individual factors, there were several scales considered: interest, efficacy, information, and professional values. Another factor considered was seniority.

As a measure for social worker's interest, we used a "discussion" scale composed of five items (family, friends, co-workers, superiors, and service users), ranging from one (never discuss social problems or policies) to four (very often discussions). The average value is three ($\alpha$ = 0.74), which reveals some degree of discussion habits and, therefore, some interest in social problems and policies.

Efficacy can be defined as the perception that individuals have of their ability to make a difference, that is, to influence the outcomes of social policies. It was measured using a 7-item scale, with values ranging between 1 and 5, and included questions about the ability to influence policies, the attention policymakers pay to social workers, and how easy it is to participate in groups or activities aimed at influencing policies. The average value is 2.99 ($\alpha$ = 0.78), indicating that efficacy is neither high nor low.

Information measures knowledge on how the policy process works. This scale was composed of five items, with values ranging between one (no knowledge at all) to five (excellent knowledge). The medium value is 3.81, which reveals some good knowledge on the policy process ($\alpha$ = 0.87).

Professional values were measured using a 7-item scale, ranging from 1 (totally disagree) to 5 (totally agree), with an average value of 2.38 ($\alpha$ = 0.687). This value, which is closer to the lower values of the scale, means that the inquired social workers have no ethical objection in engaging in policy practice.

Seniority is considered as being possibly linked to policy practice engagement; this is based on the assumption that the higher the seniority, the greater the expertise and job security to overcome some of the constraints on engagement.

A correlation analysis was performed in order to establish if there is a significant relation between policy practice engagement and the other considered variables, including discussion, efficacy, information, policy practice skills, professional values, and seniority.

Table 3 shows that, except for trust, all other considered variables show a significant relation with the level of policy practice engagement. Thus, we can say that this level increases when discussion, efficacy, information, and policy practice skills increase, and when there are no ethical constraints found.

**Table 3.** Descriptive statistics and correlations with policy practice engagement.

| | Mean | SD | N | Reliability Cronbach's $\alpha$ | Correlations (Pearson) | Correlations (Spearman) |
|---|---|---|---|---|---|---|
| PPE | 0.4732 | 0.20652 | 265 | 0.75 | 1.000 | 1.000 |
| Discussion | 3.0060 | 0.48332 | 265 | 0.74 | 0.421 ** | 0.402 ** |
| Efficacy | 2.9946 | 0.58689 | 265 | 0.78 | 0.455 ** | 0.446 ** |
| Information | 3.8143 | 0.58512 | 265 | 0.87 | 0.322 ** | 0.314 ** |
| PP skills | 3.0887 | 0.80167 | 265 | 0.83 | 0.281 ** | 0.271 ** |
| Values scale | 2.3833 | 0.63714 | 265 | 0.69 | −0.286 ** | −0.283 ** |
| Trust scale | 2.8005 | 0.56390 | 265 | 0.77 | 0.119 | 0.110 |
| Seniority | 15.15 | 10.103 | 258 | | 0.123 * | 0.125 * |

** $p < 0.01$. * $p < 0.05$

In order to understand the specific contribution of each variable to the level of policy practice engagement, we proceeded with a regression analysis. The model explains 34.3% of the differences in policy practice engagement. The main contributions come from discussion and efficacy. Information and policy practice skills are the factors that contribute the least (Table 4).

**Table 4.** Regression analysis for each predictor's contribution to explain PPE.

| Model | R | $R^2$ | Adjusted $R^2$ | Standard Error of the Estimate | Change Statistics | | | | |
|---|---|---|---|---|---|---|---|---|---|
| | | | | | $R^2$ Change | Change F | df1 | df2 | Sig. F Change |
| 1 | 0.421 [a] | 0.177 | 0.174 | 0.18772 | 0.177 | 56.519 | 1 | 263 | 0.000 |
| 2 | 0.540 [b] | 0.292 | 0.286 | 0.17445 | 0.115 | 42.538 | 1 | 262 | 0.000 |
| 3 | 0.558 [c] | 0.312 | 0.304 | 0.17234 | 0.020 | 7.472 | 1 | 261 | 0.007 |
| 4 | 0.558 [d] | 0.312 | 0.301 | 0.17263 | 0.000 | 0.120 | 1 | 260 | 0.729 |
| 5 | 0.585 [e] | 0.343 | 0.330 | 0.16903 | 0.031 | 12.185 | 1 | 259 | 0.001 |

[a]. Predictors: (Constant), Discussion; [b]. Predictors: (Constant), Discussion, Efficacy; [c]. Predictors: (Constant), Discussion, Efficacy, Information; [d]. Predictors: (Constant), Discussion, Efficacy, Information, PPskills; [e]. Predictors: (Constant), Discussion, Efficacy, Information, PPskills, Values.

## 4. Discussion

Our main conclusion points to the low level of policy practice engagement of the inquired social workers, which is in line with studies in other countries (Gal and Weiss-Gal 2015, 2013). The factors with a greater contribution on policy practice engagement are individual factors such as personal interests, as measured by social problem discussion habits, and efficacy, which is the perception of being able to make a difference.

Given that policy practice implies acting in the political structures, political participation may work as a strong predictor for engaging in policy practice. This study shows that social workers are not politically active and distrust the main political institutions. Although social workers in other countries are more politically active than the average citizen (Hamilton and Fauri 2001; Ritter 2008), the level of distrust and disaffiliation shown in this study suggest the need to further study social workers' political participation.

Professional associations and academia are the main stakeholders in the reinforcement of policy practice. Professional associations may work as actors by proxy once social workers recognise the importance of having strong professional associations that allow them to avoid direct exposure to the political structures and act "in the backstage"(Gewirtz-Meydan et al. 2016).

Regarding academic background, the results show that social workers acknowledge that influencing social policies is part of their professional purpose; however, they do not seem to know how to do so and need to develop the required skills for a better performance in this field. Therefore, academia can play a lead role in the reinforcement of social workers' activity in social policies through training, research, and expertise (Gal and Weiss-Gal and Weiss-Gal 2017, 2013; Hamilton and Fauri 2001; Hoefer 2013; Ritter 2008; Rome and Hoechstetter 2010). In this dimension, Branco (2017)'s study revealed that the most widespread policy activities that Portuguese social work academics were involved in were working with students (mean score of 2.57 out of 5) in a context of relatively low overall level of engagement in policy by the social work faculty (pp. 124–128).

These results should be interpreted with caution, given some of our study's limitations. The first limitation regards the questions in the survey. Since they were adapted from similar studies in other countries, some of the specific features of the Portuguese context may have not been fully captured within this survey, and the exclusion of other countries features that didn't apply in the Portuguese may have reduce the consistency of some of the scales. The survey recalibration would require in-depth studies, possibly of a qualitative nature, to allow for a further exploration of socio-political and professional influence on policy practice engagement.

The second limitation has to do with the unavailability of a social workers' national registry, limiting the possibilities of working with representative samples; as a consequence, our results must be considered with caution to avoid misleading generalisations.

The third limitation is an insufficient exploration of socio-political and professional factors. Our analytical model is not enough to explain policy practice engagement. Build-

ing on the Policy Practice Engagement Model (Gal and Weiss-Gal 2015) and the Civic Voluntarism Model (Verba et al. 1995), we considered social workers' engagement in policy practice activities would depend on the opportunities given by the socio-political context, the facilitation given by the professional context, the organisational setting, and individual motivation. Our results show that the individual factors are the ones which account the most for the level of policy practice engagement we could explain. Yet, the other two dimensions—socio-political context and professional setting—were underexplored in this study, and a better exploration should improve the model's capability of explaining why social workers engage or not in policy practice.

Considering the socio-political dimension, we highlight a contradiction revealed by the detachment between social workers and the political sphere, raising some concerns about the role of social workers in promoting social justice. When building our survey, we avoided the inclusion of some of the political participation questions, such as voting, engaging in campaigns, and running for elections. In Portugal, there are no different words for polity, politics, and policy, and the same word ("política") is used for all of these instances. Our results, however, illustrate concerns about the depoliticisation of the role of social workers, given the low levels of political participation and of trust in political institutions, and the high levels of dissatisfaction with the welfare state. The respondents acknowledged the connection between social work and social policymaking as a political process. The lack of identification with democratic institutions may lead to an under-exploration of the opportunities that the socio-political context may generate, widening the mismatch between decisions made and people's needs, with possibly undesirable consequences for democracy itself. Studies in other countries revealed that social workers' political participation was higher than citizens' political participation (Hamilton and Fauri 2001; Ritter 2008). Our study does not follow these studies' results, suggesting further research to specifically address social workers' political participation.

The professional context, in our study, was not found to be relevant for explaining policy practice engagement. This may also be related to the insufficient adaptation of the questions about the professional context to the Portuguese case. Our results show a gap between discourses and practices, given the high levels of importance given to professional organisations despite the low levels of participation in such organisations. Further research is needed since engagement in professional associations has been pointed out as a driver for policy practice engagement (Gal and Weiss-Gal 2015; Hamilton and Fauri 2001). Similar to what Guidi (2020) shows for the Italian and Spanish cases, the approval by the Portuguese Parliament of the Law that creates the Portuguese College of Social Works (Law nº 121/2019) could be an opportunity to change this scenario. Firstly, because as legal public entities, social work professional organisations assume a role of institutionalised partners on public policies and professional training education and "are therefore [. . .] able to represent and defend the professional values that put users' rights and social justice centre stage" (p. 1052). Secondly, the accreditation and regulation powers of these organisations have the potential to broaden their professional representativity since enrolment is mandatory to work as a licensed social worker. Additionally, the funding, derived from members fees, can be an important resource to support stronger professional organisations and their capacity for collective action. However, how such representativity will increase the ability to influence social policy, both through collective action and individual practice, will require further study, as underlined by Guidi (2020).

## 5. Conclusions

In this study, we aimed to understand if Portuguese social workers engage in policy practice activities and which factors enable or constraint such practice. Our results show that there is a low level of policy practice engagement by Portuguese social workers. The main facilitators for engaging are individual factors, such as policy discussion in personal or professional networks, and self-perception of the ability to make a difference (efficacy).

This study provided insights about the reinforcement of social workers' critical self-reflection and the relevance of finding strategies to empower social workers to develop alternatives to the top-down relation between social policies and social workers, reinforcing their policy practice.

Based on our survey results, we can say that the inquired social workers show a low level of policy practice engagement, despite the reported unsatisfaction with the welfare system. They also acknowledge that it is within social workers' mandate to engage in social policy, even though they do not engage.

Our results are not generalisable since we did not work with a representative sample. Even so, we can observe some trends which challenge us to look for strategies that focused on enhancing social workers' motivation and skills in policy practice.

Being the first extensive study on policy practice in Portugal, this study is only a starting point in this research area. It expands our knowledge on how the relation between social work and social policy in Portugal works, and it points out some perspectives that, if taken in consideration, can help to ensure that social workers are better prepared to recentre social policies in people's needs, fulfilling their social mission. Such strategy can bring professional benefits, but mostly societal benefits.

Since there is a preference for acting in the "backstage", professional associations may work as actors by proxy helping to avoid social workers' direct exposure to political structures.

Academia can play a key role in teaching and raising policy practice skills, expanding the knowledge on social problems and policies through research, and participating in the policy process with academic expertise.

Considering that the respondents are concerned with their professional development, as evident in their postgraduation qualification, incorporating content related to the ability to influence social policies into postgraduate academic curricula may be a way to increase this professional practice.

This research field is underexplored in Portugal, and our study confirms the need to further this line of work, especially in current times, where the welfare state is being challenged by multiple and successive crises.

**Author Contributions:** The present article has, as a base, the master thesis in Social Work of the R.C. with the supervision of F.B. In this larger work, both authors contributed to both the design of this research and the present article to be published. The R.C. had a major contribution to data collection, first-level analysis of the empirical data and the draft of the article. The F.B. had a significant contribution to data analysis revision, interpretation and critical revision and development of the paper manuscript. All authors have read and agreed to the published version of the manuscript.

**Funding:** This research received no external funding.

**Institutional Review Board Statement:** Ethical review and approval were waived for this study due to the fact that the participation in the study was risk-free for participants, was voluntary and informed.

**Informed Consent Statement:** Human participants consent was assured considering that the method of the data collection (voluntary participation on online survey) implies an informed consent. The first section of the online questionnaire informed the participants about is compliance with to the ethical research standards, namely the anonymization and the safeguard of the confidentiality of responses, before the answer to any question.

**Data Availability Statement:** Data presented in this study are available in institutional repository [http://hdl.handle.net/10400.14/25740].

**Acknowledgments:** The authors acknowledge the collaboration of the Portuguese Association of Social Workers—APSS in the collaboration in the dissemination of the survey among its members and network.

**Conflicts of Interest:** The authors declare no conflict of interest.

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
