# Peer review of "Social Workers’ Involvement in Policy Practice in Portugal"

_socsci, doi:10.3390/socsci12020105_

Round 1
Reviewer 1 Report
This paper highlights how social workers influence socal policy, using Verba's Civic Voluntarism Model, and Gal & Weiss-Gal Policy Practice Engagement Model. It is a quantitative study that used the snowball method to access Social Workers
In the introduction it is assumed that Social Work has a relationship with public policies, being influenced and influencing them, and therefore, the practice developed by professionals in this area is socio-political.
It is necessary to clarify this relationship because socio-political practice is not the same as practicing in social policies. Therefore, I ask what is the relevance of addressing the specificity of the political practice of social work in the context of public policies. As we know, socio-political practice has a specificity, and this specificity must be made clear: either it is reflected upon and deepened, or else it will have to be removed from the reflection. The introduction also assumes that this is the first extensive study of this nature in Portugal (the word extensive is used several times in the article without any methodological theoretical explanation to that effect). But what does extensive mean? Does it mean that it was applied to a considerable number of social workers? Or is it just because it is a quantitative study? If so, it is necessary to justify whether the social workers who answered the questionnaire are from which region of the country, and whether they are representative in national terms, as well as in terms of the practice of the profession, which policy areas they practice in, as this is what the article is about.
The introduction also points out some results, but has not yet explained how it arrived at them, so this part is meaningless. That said, the introduction needs a revision, focusing on the focus of the article.
In the conceptual framework and methods, it is important to mention that the theoretical models adopted are explained, but without any definition of concepts or problematisation of them, including lack of a clear literature review on the topic. this is a serious problem in the article.
As for the methodology, the issue that is of most concern is the application of this online questionnaire, without any control over who responds to them, which may be social workers or other professionals. What control mechanism was used in this case for the study? On the other hand, the way it was applied is not sufficiently clear. It is also worth mentioning that the data was collected in 2017, which already makes up almost 6 years since its collection (however in the tables in the following point the data presented is from 2018).
The results are presented revealing the data discovered with the application of the questionnaire. The discussion of the results needs to be arranged by means of the findings of the questionnaire and only then present the limits and potentialities of the study.
The conclusion lacks depth in answering the starting question and the objectives of the study.
It should be noted that the number of bibliographical references presented is very limited and there are at least two repeated references. There are hundreds of articles on the subject that should enable and help to deepen the conclusions of the study. Although the topic is interesting it does not meet the requirements for a coherent article on the subject.
Author Response
We would like to thank you for your revision of our article and the criticism and suggestions you made.
Trying to answer to your comments we would like to register the following.
- In our revision of the article we try to make clearer the relationship social work – public policies and, also exemplifying different interventions of social workers in the policy cycle both at the street-level and in other stages of the policy cycle. (Please see section 1)
- We reinforce the component of literature review namely trying to contextualise the social work policy practice as conceptualised in our research facing the several lines of research about the more broaden field of social work policy engagement in the last two decades. (Please see namely section 1)
- We try to be more precise about the nature and features of our empirical research aiming to eliminate possible methodological ambiguities (our research is a survey-based research, developed in the absence of Portuguese social work directory (PSW), trying to cover a representative sample of PSW in terms of sample dimension, the SW main sectors and regions of activity, and adopting validation procedures namely regarding the control mechanism of respondents as licensed social workers. (Please see section 2)
- We try to explain better the conceptual framework (Please see section 2)
- The piece of research on which the article is based was developed between the last trimester of 2017 (data gathering) and the first semester of 2018 (data validation and analysis). We considered the data analysis (2018) following a practice used by some official statistical agencies.
We would like to say that this piece of research was presented in two conferences, one of them recently, and we are encouraged to publish it considering the lack of publication about this topic in Portugal. It is for this reason that we consider the publication despite the time elapsed since the empirical and the analytical work.
Reviewer 2 Report
The article starts by claiming that social workers should be involved in influencing social policy (making). It refers to the international professional discourse on policy practice and focuses on the case of Portugal.
The conceptual model merges the Civic Voluntarism Model and the Policy Practice Engagement Model. The descripton of the conceptual model is very basic and should be explained in more detail. Which parts come from which Model-Approaches? And please explain figure 1 in the text.
The method of data collection as well as the sample is described in full detail.
The results focus on
-the general relationship between social work and social policy which reveals a lack of skills in policy practice (skills are also mentioned below: where ist the place to put them?),
- policy practice activities which show that social workers avoid public exposure,
- the low level of policy practice activities (this subchapter could be merged with the aforementioned),
- factors that enhance or constrain policy practice (page 8/line 326: seniority should also be included in the text), finding that individual factors hold the most potential for explaining the low level of policy practice activities.
The discussion is coherent and gives important reflections on the limitations of the study. It also reveals the importance of professional associations and academia. The role of professional associations should be explained in more detail. This is an interesting point and could also be linked to research by Riccardo Guidi on Italy and Spain.
All in all the paper is an important contribution to the literature on policy practice of social workers. It is the first survey for Portugal and thus an important piece of knowledge. Nevertheless, some points can be improved. There is also some redundancy in the paper (e.g. between discussion and exclusion, also between chapter 3.2 and chapter 3.2.3).
Author Response
We would like to thank you for your revision of our article and the criticism and suggestions you made.
- Trying to answer to your comments we reviewed the description of the conceptual framework in a more detailed manner (Please see section 2)
- Regarding the respondents’ seniority we considered it in our correlational analysis (Please see Table 3)
- We welcome the suggestion of to consider the role of professional associations as one of the routes of professional policy engagement. Then we referred to this analytical lens in our discussion/conclusion.
Reviewer 3 Report
This is an important study as it is the first quantitative study on policy practice undertaken in Portugal and, as such, sheds original and relevant light on the subject, which has been attracting growing attention in the literature. I have some suggestions for addressing some issues in the text:
1. While the English is generally good, I think that the article could use additional language editing.
2. On page. 1, you describe the role of social workers in implementation. The term "discretion" could be useful here.
3. In the first para. on p. 3, I suggest adding some sources to the claim that PP is gaining additional interest in the discourse - Gal & Weiss-Gal, 2023; De Corte & Roose, 2020; Pawal & Nixon, 2021; Chandrapal, 2022.
4. Some more details on the data collection would be useful.
5. The terms in some of the tables are a bit unclear. What does "relate social work to social policy formulation" imply exactly (T.2)?
6. The last para. on p. 7 is not clear.
7. The sentence pertaining to the role of professional associations on p. 9 is not clear as this issue did not seem to be central in the analysis.
Author Response
We would like to thank you for your revision of our article and the criticism and suggestions you made.
- In our revision of the article we try to make clearer the relationship social work – public policies and, also to clarify different dimension of the possible fields of intervention of social workers in the policy cycle both at the street-level and in other stages of the policy cycle. In this context we underlined the mobilise the social work “discretion” as one of the relevant dimensions on policy practice analysis (Please see section 1)
- We reinforce the component of literature review namely trying to contextualise the social work policy practice as conceptualised in our research facing the several lines of research about the more broaden field of social work policy engagement in the last two decades. (Please see namely section 1)
- We try to be more precise about the nature and features of our empirical research aiming to eliminate possible methodological ambiguities (our research is a survey-based research, developed in the absence of Portuguese social work directory (PSW), trying to cover a representative sample of PSW in terms of sample dimension, the SW main sectors and regions of activity, and adopting validation procedures namely regarding the control mechanism of respondents as licensed social workers. (Please see section 2)
- Categories of T2 was revised
- We welcome the suggestion of to consider the role of professional associations as one of the routes of professional policy engagement. Then we referred to this analytical lens in our discussion/conclusion.
Round 2
Reviewer 1 Report
The article reveals that the authors responded to the reviewers' topics.
In general there were substantial improvements especially in the revision of the bibliography, increase in the number of references, and clarification of the methodology.
The article can be published but first it has to be proofread in English.
On the other hand, it would be recommended that the discussion be substantiated with the authors of the literature review.
The conclusion can be shorter, mainly by reducing the repetitions already illustrated in the discussion of the results.
But authors feel free to object, I have no objections to the publication of the article.
With these changes, the article turns out to be an excellent contribution to social work in Portugal.
Author Response
Thank you again for your comments and suggestions. We tried to improve the discussion section introducing, basically, some of the relevant references.
Regarding the conclusion we didn't change it because we considere the is difficult to avoid some redundancy given the proposal to summarise some of the more relevant aspects of the article.
Regarding the English proofing we will check with journal editor if it will be absolutely necessary.
Reviewer 2 Report
Maybe you would like to add another reference on page 3, first paragraph at the end:
Leitner, Sigrid and Klaus Stolz (2022): German Social Workers as Professional Politicians: Career Paths and Social Advocacy, in: European Journal of Social Work, published online 31.08.2022, https://doi.org/10.1080/13691457.2022.2117138
Author Response
Thank your for your suggestion that we omitted involuntary in the 1st revision but it is included in this version.